# Outcome of COVID-19 in hospitalised immunocompromised patients: An analysis of the WHO ISARIC CCP-UK prospective cohort study

**Lance Turtle**[1,2☯]*, **Mathew Thorpe**[3☯], **Thomas M. Drake**[3], **Maaike Swets**[4,5], **Carlo Palmieri**[6], **Clark D. Russell**[7], **Antonia Ho**[8], **Stephen Aston**[2,9¤], **Daniel G. Wootton**[1,2], **Alex Richter**[10,11], **Thushan I. de Silva**[12], **Hayley E. Hardwick**[1], **Gary Leeming**[13], **Andy Law**[5], **Peter J. M. Openshaw**[14], **Ewen M. Harrison**[3], **ISARIC4C investigators**[¶], **J. Kenneth Baillie**[5,15,16‡], **Malcolm G. Semple**[1,17‡], **Annemarie B. Docherty**[3,15‡]

1 NIHR Health Protection Research Unit in Emerging and Zoonotic Infections, Department of Clinical Infection, Microbiology and Immunology, Institute of Infection, Veterinary and Ecological Sciences, University of Liverpool, Liverpool, United Kingdom, 2 Liverpool University Hospitals NHS Foundation Trust, Liverpool, United Kingdom, 3 Centre for Medical Informatics, Usher Institute, University of Edinburgh, Edinburgh, United Kingdom, 4 Department of Infectious Diseases, Leiden University Medical Centre, Leiden University, Leiden, the Netherlands, 5 The Roslin Institute, Easter Bush campus, University of Edinburgh, Edinburgh, United Kingdom, 6 Department of Molecular and Clinical Cancer Medicine, Institute of Systems, Molecular and Integrative Biology, University of Liverpool, Liverpool, United Kingdom, 7 University of Edinburgh Centre for Inflammation Research, Queen's Medical Research Institute, Edinburgh, United Kingdom, 8 MRC-University of Glasgow Centre for Virus Research, University of Glasgow, Glasgow, United Kingdom, 9 Institute of Infection, Veterinary and Ecological Sciences, University of Liverpool, Liverpool, United Kingdom, 10 Institute of Cancer and Genomic Science, College of Medical and Dental Science, University of Birmingham, Birmingham, United Kingdom, 11 University Hospitals Birmingham NHS Foundation Trust, Birmingham, United Kingdom, 12 Department of Infection, Immunity and Cardiovascular Disease, Medical School, The University of Sheffield, Sheffield, United Kingdom, 13 Department of Biostatistics, University of Liverpool, Liverpool, United Kingdom, 14 National Heart and Lung Institute, Imperial College London, London, United Kingdom, 15 Intensive Care Unit, Royal Infirmary Edinburgh, Edinburgh, United Kingdom, 16 Baillie Gifford Pandemic Science Hub, Centre for Inflammation Research, University of Edinburgh, Edinburgh, United Kingdom, 17 Respiratory Medicine, Alder Hey Children's Hospital, Liverpool, United Kingdom

☯ These authors contributed equally to this work.
‡ These authors are joint senior authors on this work.
¤ Current address: Institute of Systems, Molecular and Integrative Biology, University of Liverpool, Liverpool, United Kingdom.
¶ Membership of ISARIC4C investigators is provided in S1 Acknowledgments.
* lance.turtle@liverpool.ac.uk

**Data Availability Statement:** This work uses data provided by patients and collected by the NHS as part of their care and support #DataSavesLives. Data cannot be shared publicly because they contain potentially identifiable information. The CO-CIN data were collated by ISARIC4C Investigators.

## Abstract

### Background

Immunocompromised patients may be at higher risk of mortality if hospitalised with Coronavirus Disease 2019 (COVID-19) compared with immunocompetent patients. However, previous studies have been contradictory. We aimed to determine whether immunocompromised patients were at greater risk of in-hospital death and how this risk changed over the pandemic.

### Methods and findings

We included patients > = 19 years with symptomatic community-acquired COVID-19 recruited to the ISARIC WHO Clinical Characterisation Protocol UK prospective cohort

ISARIC4C welcomes applications for data and material access through our Independent Data and Material Access Committee (https://isaric4c.net). NHS Digital data can be access by application to the NHS Digital Data Access Request Service (DARS) at https://digital.nhs.uk/services/data-access-request-service-dars.

**Funding:** This work is supported by the National Institute for Health Research (NIHR, https://www.nihr.ac.uk) [award CO-CIN-01 to MGS, Senior Investigator Award 201385 to PJMO and Advanced Fellowship NIHR300669 to DGW]. This work is supported by the Medical Research Council (MRC, https://www.ukri.org/councils/mrc/) [grant MC_PC_19059 to JKB, MGS, and PJMO and MR/V028979/1 to LT and CP], the Chief Scientist Office, Scotland (https://www.cso.scot.nhs.uk) [to ABD, no ref number]. This work is supported by the Wellcome Trust (www.wellcome.ac.uk) [215091/Z/18/Z to MGS and JKB (with the Department for International Development), 205228/Z/16/Z to LT]. This work is supported by the Bill and Melinda Gates Foundation [OPP1209135 to MGS and JKB]. This work is supported by the Liverpool Experimental Cancer Medicine Centre [C18616/A25153 to CP and MGS]. LT, MGS and HH are also supported by the NIHR Health Protection Research Unit (HPRU) in Emerging and Zoonotic Infections at University of Liverpool in partnership with the UK Health Security Agency (UK-HSA), in collaboration with Liverpool School of Tropical Medicine and the University of Oxford [award 200907]. PJMO is also supported by the NIHR HPRU in Respiratory Infections at Imperial College London with UK-HSA [award 200927], the NIHR Biomedical Research Centre at Imperial College London [IS-BRC-1215-20013], and the EU Platform foR European Preparedness Against (Re-) emerging Epidemics (PREPARE) [FP7 project 602525]. This research is part of the Data and Connectivity National Core Study, led by Health Data Research UK in partnership with the Office for National Statistics and funded by UK Research and Innovation [MC_PC_20029]. The funders had no role in study design, data collection and analysis, decision to publish, or preparation of the manuscript.

**Competing interests:** I have read the journal's policy and the authors of this manuscript have the following competing interests: LT declares a lecture fee paid to his institution from Eisai ltd. CP reports research grants from Pfizer, Daiichi Sankyo and Seagen, consulting fees from Pfizer, Roche, Daiichi Sankyo, Novartis, Exact sciences, Gilead, SeaGen and Eli Lilly, and payment for lectures from Pfizer, Novartis and Eisai ltd. PO declares fees from

study. We defined immunocompromise as immunosuppressant medication preadmission, cancer treatment, organ transplant, HIV, or congenital immunodeficiency. We used logistic regression to compare the risk of death in both groups, adjusting for age, sex, deprivation, ethnicity, vaccination, and comorbidities. We used Bayesian logistic regression to explore mortality over time. Between 17 January 2020 and 28 February 2022, we recruited 156,552 eligible patients, of whom 21,954 (14%) were immunocompromised. In total, 29% ($n = 6,499$) of immunocompromised and 21% ($n = 28,608$) of immunocompetent patients died in hospital. The odds of in-hospital mortality were elevated for immunocompromised patients (adjusted OR 1.44, 95% CI [1.39, 1.50], $p < 0.001$). Not all immunocompromising conditions had the same risk, for example, patients on active cancer treatment were less likely to have their care escalated to intensive care (adjusted OR 0.77, 95% CI [0.7, 0.85], $p < 0.001$) or ventilation (adjusted OR 0.65, 95% CI [0.56, 0.76], $p < 0.001$). However, cancer patients were more likely to die (adjusted OR 2.0, 95% CI [1.87, 2.15], $p < 0.001$). Analyses were adjusted for age, sex, socioeconomic deprivation, comorbidities, and vaccination status. As the pandemic progressed, in-hospital mortality reduced more slowly for immunocompromised patients than for immunocompetent patients. This was particularly evident with increasing age: the probability of the reduction in hospital mortality being less for immunocompromised patients aged 50 to 69 years was 88% for men and 83% for women, and for those >80 years was 99% for men and 98% for women. The study is limited by a lack of detailed drug data prior to admission, including steroid doses, meaning that we may have incorrectly categorised some immunocompromised patients as immunocompetent.

## Conclusions

Immunocompromised patients remain at elevated risk of death from COVID-19. Targeted measures such as additional vaccine doses, monoclonal antibodies, and nonpharmaceutical preventive interventions should be continually encouraged for this patient group.

## Trial registration

ISRCTN 66726260.

## Author summary

### Why was this study done?

- Throughout the Coronavirus Disease 2019 (COVID-19) pandemic, mortality has been much higher in certain groups of patients. Older people and those with underlying medical conditions have been at greater risk of death.

- Over time, with improvements in medical care and especially vaccination, mortality from COVID-19 has reduced dramatically.

- Patients with a weakened immune system are also at higher risk from COVID-19. It is not clear whether there is an increased risk of needing admission to hospital, or whether once immunocompromised patients are in hospital their risk of death is also increased, compared with immunocompetent patients.

Affnivax, Oxford Immunotech, Nestle, Pfizer and Janessen for consulting or chairing a symposium, paid to Imperial College. MGS is chair of the infectious disease advisory board for Integrum Scientific, director of MedEx solutions ltd, participated in the DSMB for Pfizer COVID-19 vaccine trials, and was donated an investigational medicinal product by Chiesi Farmaceutici S.p.A. MGS and PJMO sat on HMG UK New Emerging Respiratory Virus Threats Advisory Group (NERVTAG). MGS sat on HMG UK Scientific Advisory Group for Emergencies (SAGE), COVID-19 Response from March 2020 to March 2022.

**Abbreviations:** CCP, Clinical Characterisation Protocol; COVID-19, Coronavirus Disease 2019; CRP, C-reactive protein; IL-6, interleukin 6; IQR, interquartile range; ISARIC, International Severe Acute Respiratory and emerging Infection Consortium; LLSOA, lower layer super output area; NIMS, national immunisation management system; ONS, Office of National Statistics; OR, odds ratio; REDCap, Research Electronic Data Capture; RT-PCR, reverse transcriptase polymerase chain reaction; SDC, statistical disclosure control.

## What did the researchers do and find?

- We sought to determine whether the risk of death was higher in immunocompromised patients after admission to hospital, using the WHO International Severe Acute Respiratory and emerging Infection Consortium (ISARIC) Clinical Characterisation Protocol (CCP) UK dataset, a prospective observational cohort study of hospitalised patients in the United Kingdom.

- Our analysis showed that patients who are immunocompromised have a higher risk of death in hospital than patients with intact immune systems. This difference remained even accounting for other important factors such as age, sex, and the presence of chronic conditions.

- Over the course of the pandemic, although the risk of death for all patients has decreased, the risk has decreased much more for immunocompetent patients and the gap has widened for immunocompromised patients.

## What do these findings mean?

- Immunocompromised patients remain at increased risk of death compared with other patients admitted with symptomatic COVID-19.

- Clinicians and policy makers should be aware of the increased risk of death in this patient group.

- Targeted interventions such as antiviral treatments, antibodies, and nonpharmaceutical interventions should continue to be used in this patient group.

## Introduction

Coronavirus Disease 2019 (COVID-19), the disease caused by Severe Acute Respiratory Syndrome Coronavirus 2 (SARS-CoV-2), disproportionately affects older people and those with underlying health conditions [1]. However, a key challenge throughout the pandemic has been to delineate which comorbidities confer the greatest risk. Severe COVID-19 is an inflammatory process [2,3], with several lines of evidence suggesting that immune dysfunction is linked to adverse outcome [4,5]. Consistent with this are clinical trial findings that anti-inflammatory treatments with dexamethasone, tocilizumab, and baricitinib improve survival in patients with respiratory failure [6–8].

Early data in the pandemic suggested a high mortality among immunocompromised patients, with organ transplant recipients being at particular risk [9,10]. Studies have compared mortality among immunocompromised patients with other patient groups with COVID-19 with conflicting results. Some studies show increased mortality [11,12], whereas others show no difference from other patient groups [13,14]. A challenge when attributing risk to individual comorbidities has been that many factors are correlated. For example, some kidney transplant recipients also have abnormal renal function, which is itself a risk factor for poor outcome from COVID-19 [1,15]. A large UK population study using routine health data from 17 million primary care health records (OPENSafely) found that immunocompromising conditions, including organ transplant and haematological malignancy, increased the risk of

COVID-19-associated death [16]. However, detailed information on disease severity at presentation and events during hospitalisation were not available.

As public health measures to control the spread of SARS-CoV-2 have now eased, concern remains for the safety of those who are immunocompromised and whose vaccination response may be compromised [17]. Our aim was therefore to analyse one of the largest prospective cohorts of hospitalised COVID-19 cases, the International Severe Acute Respiratory and emerging Infection Consortium (ISARIC) WHO Clinical Characterisation Protocol in the United Kingdom (CCP-UK) study dataset, to test the hypothesis that outcomes are worse in immunocompromised patients, and whether the improved survival observed over the course of the pandemic was reduced in immunocompromised patients.

## Methods

### Study design and setting

The ISARIC WHO CCP-UK prospective observational cohort study was activated on 17 January 2020 as part of the public health response to the COVID-19 pandemic. CCP-UK prospectively recruited a cohort of >300,000 patients, hospitalised with COVID-19, from 306 healthcare facilities across the UK. The sample size was not prespecified. The protocol, revision history, case report forms, and consent forms are available online at isaric4c.net. The study received ethical approval from the South Central—Oxford C Research Ethics Committee in England (Ref: 13/SC/0149) and by the Scotland A Research Ethics Committee (Ref: 20/SS/ 0028). This study is reported according to the Strengthening the Reporting of Observational Studies in Epidemiology (STROBE) guidelines for cohort studies (see S1 STROBE Checklist).

### Participants

Adults (≥19 years) who were admitted to hospital between 17 January 2020 and 28 February 2022 with confirmed or highly suspected SARS-CoV-2 infection leading to COVID-19 were included in this analysis. During the first wave, highly suspected cases were also eligible for inclusion, because SARS-CoV-2 was an emergent pathogen at that time and laboratory confirmation was dependent on availability of testing. SARS-CoV-2 infection was confirmed using reverse transcriptase polymerase chain reaction (RT-PCR). Participants 18 years and under were excluded because these cases had been previously reported [18].

### Data sources

Data collected by research staff were entered into a standardised electronic case report form within a secure Research Electronic Data Capture (REDCap) database. Vaccination data were obtained from the national immunisation management system (NIMS) and deterministically linked to the ISARIC CCP-UK REDCap data using NHS number, which was collected as part of the ISARIC CCP-UK dataset. Vaccination data were not available for Scottish patients, and these analyses are therefore restricted to only English and Welsh ISARIC CCP-UK participants with a valid, linkable NHS number. Participants with invalid NHS numbers were excluded from the analysis after December 2020 to allow accurate linkage with vaccination data. Readmissions and those with erroneously recorded admission dates (e.g., those with admission dates outside the scope of the study) were also removed prior to analysis. The final time point data of analysis was 28 days after admission. Patients were considered to be alive if the 28-day outcome was missing. There are no data used after day 28 of follow-up.

## Variables

**Immunocompromise.** We considered patients to be immunocompromised if they met any of the following clinical criteria: solid organ transplant, active cancer diagnosis and treatment, congenital immune deficiency, human immunodeficiency (HIV) infection, in receipt of preadmission immune-suppressive treatments, or preadmission oral or intravenous steroids. For the purposes of analysis, these patients were only considered to be in one category, with the underlying reason for being immunocompromised being categorised by the following hierarchy: inherited immunological or metabolic disorder > solid organ transplant > cancer > HIV > preadmission immunosuppressants > preadmission steroids. Overlap between immunocompromising factors were visualised as part of the analysis.

**Participant characteristics.** Patient demographics including age, sex, comorbidities, and ethnicity were recorded at hospital admission. Deprivation index was calculated using lower layer super output area (LLSOA) data provided by the Office of National Statistics (ONS). Physiological parameters of the 4C Mortality Score [19] were used as markers of illness severity at presentation: respiratory rate (breaths per min), Glasgow Coma Scale, oxygen saturation (%), blood urea (mmol/L), blood C-reactive protein (CRP, mg/L). Symptoms recorded on admission are listed in the CRF, which is available at isaric4c.net. If no symptom criteria were met, a patient was considered asymptomatic and was excluded from the analysis. If a patient met at least one of the criteria, they were considered to be symptomatic and were included.

In-hospital interventions were recorded including critical care admission, level of respiratory support, and treatments for COVID-19 including corticosteroids and interleukin 6 (IL-6) receptor blockers. For steroid and anti-IL-6 treatments, analyses were restricted to patients on oxygen therapy. Treatment with IL-6 receptor blockers was indicated by national guidance for patients with a CRP blood level of 75 mg/l or greater, or on respiratory support. Due to the nature of the data collection, preadmission steroids were assessed through a free text search of preadmission medication; dexamethasone, beclometasone, prednisolone, cortisone/hydrocortisone, or betamethasone were included.

**Vaccination.** To investigate the effect of vaccination, we considered patients having received no vaccine doses or within 20 days of the first vaccine dose as being unprotected and therefore having no immunity to SARS-CoV-2; we then stratified patients as having received 1, 2, 3, and 4 or more doses, provided 3 weeks or more had elapsed between the first vaccination and symptom onset or positive RT-PCR test (whichever was earlier), or 1 week for subsequent doses, to allow immunity to develop [20]. Vaccination status was incorporated into the analysis as a multilevel variable.

**Pandemic waves and SARS-CoV-2 variants.** Wave 1 was considered to be between 17 January 2020, the date that the study protocol was activated in the UK, and 31 August 2020, the nadir of hospital inpatient numbers between the first and second waves. This wave was largely accounted for by the B.1 lineage SARS-CoV-2 ancestral strain containing the D614G mutation. Wave 2 was defined as the period from 1 September 2020 to 31 March 2021 and comprised a mixture largely of B.1 D614G lineages and the alpha variant (B.1.1.7). The third wave, during which the Alpha variant was replaced by the Delta variant (B.1.617.2), was between 1 April 2021 and 12 December 2021. From 13 December 2021 onwards, Omicron (B.1.1.529) became the dominant circulating variant in the UK and outcompeted Delta [21]. We refer to this period as the fourth wave.

**Outcomes.** The primary outcome was in-hospital mortality. Secondary outcomes were the use of oxygen, noninvasive ventilation, invasive mechanical ventilation, and admission to critical care.

## Missing data

Due to the nature of such a large-scale, observational study conducted during pandemic surge conditions, high degrees of missingness exist in multiple variables, particularly in the most recent wave dominated by the Omicron variant where recruitment has waned over time. A sensitivity analysis using data imputation was performed to assess the impact of missingness.

## Statistical analyses

Continuous data are summarised as median (interquartile range (IQR)) and categorical data as frequency (percentage). We employed statistical disclosure control (SDC) measures to protect patient confidentiality and anonymity.

To examine in-hospital mortality, we initially performed multivariable logistic regression adjusted for age, sex, number of comorbidities, ethnicity, deprivation index, and vaccination [1]. The number of comorbidities did not include the condition giving rise to immunocompromise, but rather counted the number of comorbidities over and above this condition. These variables were selected for inclusion in models based on their important effects previously described by ourselves and others [1,22]. Other outcome measures were the use of oxygen, noninvasive ventilation, intensive care admission, and invasive ventilation and were adjusted for the same variables. We subsequently explored whether changes in in-hospital mortality over time were different between immunocompetent and immunocompromised patients. To allow probabilistic interpretations of absolute risk differences, a Bayesian logistic regression model was specified with weakly informative priors on model coefficients (four chains, 500 warmup, 2,000 iterations). The probability of death was determined for immunocompromised and immunocompetent patients in the first wave and the fourth wave, accounting for age, sex, socioeconomic deprivation, and comorbidity count. We then calculated the absolute risk difference for mortality in immunocompromised and immunocompetent patients in the first wave and compared this with the absolute risk difference in the Omicron wave. We set comorbidity count to "2+" and deprivation score to "2", the most common level for each variable, and stratified these results by sex, age category, and vaccination status. All analysis was performed using the statistical software package R version 4.1.1 including the use of the associated packages tidyverse [23], finalfit [24], brms [25], and UpSetR [26].

## Results

Between 17 January 2020 and 28 February 2022, data for 304,628 admissions of all ages were collected. Outcome data were available for 156,552 unique adult index admissions with symptomatic COVID-19 and valid NHS numbers (Fig 1). A total of 134,598 (86%) patients were classified as immunocompetent, and 21,954 (14%) were immunocompromised (Fig 1). There was overlap between the immunocompromising conditions (Fig 2A), for example, a large proportion of patients with previous solid organ transplant were also taking immunosuppressive medication. Using the hierarchical categorisation defined above, most patients that were identified as immunocompromised were either those taking immunosuppressive medication with no other documented immunocompromise ($n$ = 12,701 of 21,954, 58%), or those who had received recent cancer treatment ($n$ = 5,116, 23%). The numbers of patients with other conditions were as follows: inherited immune deficiency ($n$ = 526, 2%), previous solid organ transplant (1,559, 7%), HIV/AIDS (498, 2%), and preadmission steroids ($n$ = 1,554, 7%).

The median age of the immunocompetent patients was 69.5 years (IQR 53.4 to 82.0) and was slightly younger than that of the immunocompromised group, which was 71.5 (IQR 58.7 to 80.3) (Table 1). Ages of immunocompromised patients varied depending on the aetiology of their immunocompromise. Those with cancer (median age 72.0. IQR [62.0 to 79.7]) and

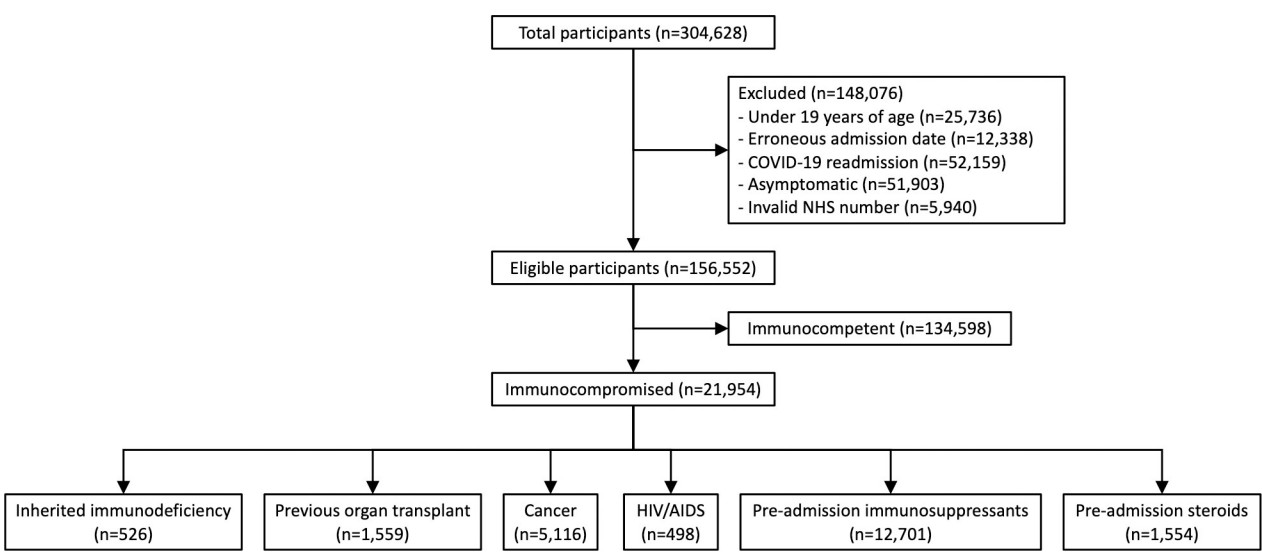

**Fig 1. Flow diagram of patients in the study.** The total number of participants entered into the ISARIC CCP-UK database, reasons for exclusion, and numbers of patients in each category of immunocompromise are shown. CCP-UK, Clinical Characterisation Protocol in the United Kingdom; COVID-19, Coronavirus Disease 2019; ISARIC, International Severe Acute Respiratory and emerging Infection Consortium.

those on preadmission immunosuppressants (median 72.5 [IQR 59.1 to 81.2]) or corticosteroids (75.1 [63.0 to 83.6]) were considerably older than those with inherited immune deficiency (median 61.8 [IQR 45.3 to 77.2]), solid organ transplant (median 62.4 [IQR 52.1 to 72.0]), and HIV (57.8 [IQR 49.8 to 70.5]). The majority, 95% (*n* = 20,801 of 21,954) of patients with immunocompromise had at least one additional comorbidity (the reason for immunocompromise was not included in the comorbidity count). Chronic pulmonary disease (*n* = 6,190 of 21,954, [28%], immunocompromised) versus (*n* = 18,495 of 134,598, (14%), immunocompetent), haematologic disease (*n* = 2,041 [9%]) versus (*n* = 3,665 [2%]), rheumatologic disease (*n* = 4,540 [21%]) versus (*n* = 12,901 [10%]), kidney disease (*n* = 4,266 [19%]) versus (*n* = 18,901 [14%]), and malignant neoplasm (*n* = 5,113 [23%]) versus (*n* = 8,484 [6%]) were more common in immunocompromised versus immunocompetent patients. Diabetes (*n* = 6,042 [28%] immunocompromised) versus (*n* = 36,322 [27%] immunocompetent), obesity (*n* = 2,945 [13%]) versus (*n* = 17,462 [13%]), hypertension (*n* = 9,758 [44%]) versus (*n* = 57,117 [42%]), chronic cardiac disease (*n* = 6,607 [30%]) versus (*n* = 35,597 [26%]) were similar in both groups (Table 1). The characteristics of immunocompetent versus immunocompromised patients according to pandemic waves are shown in S1 Table.

Overall, 114,364 of 134,598 (85%) immunocompetent and 17,737 of 21,954 (81%) immunocompromised patients hospitalised with COVID-19 were unvaccinated—nearly half the patients (76,948 of 156,552 [49%]) were admitted before vaccines were available in waves 1 and 2 (Table 1 and S1 Fig). A total of 5,112 (4%) of immunocompetent and 789 (4%) of immunocompromised patients admitted had received one dose of vaccine >20 days earlier at the time of hospital admission. A total of 11,601 (9%) immunocompetent and 2,804 (13%) immunocompromised patients had received two doses >7 days earlier. A total of 592 (<1%) immunocompetent and 290 (1%) immunocompromised patients had received 3 doses >7 days before the date of hospital admission (Table 1). Over time, the relative proportion of immunocompromised patients who were unvaccinated was less than for immunocompetent patients (S2 Fig).

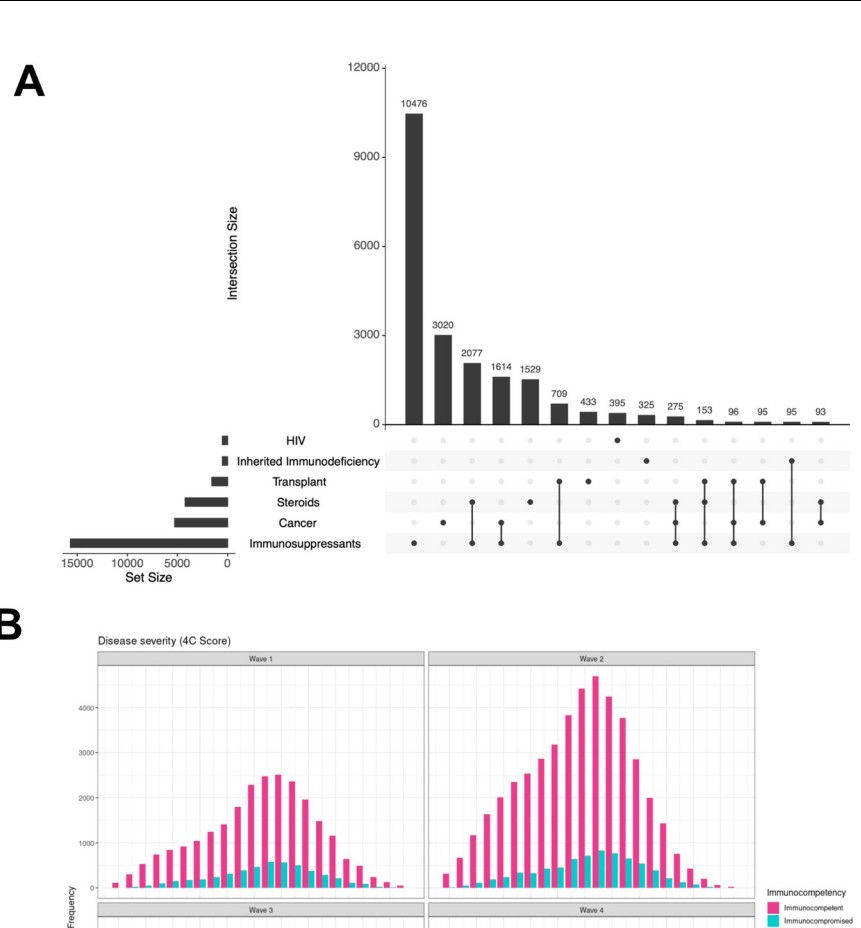

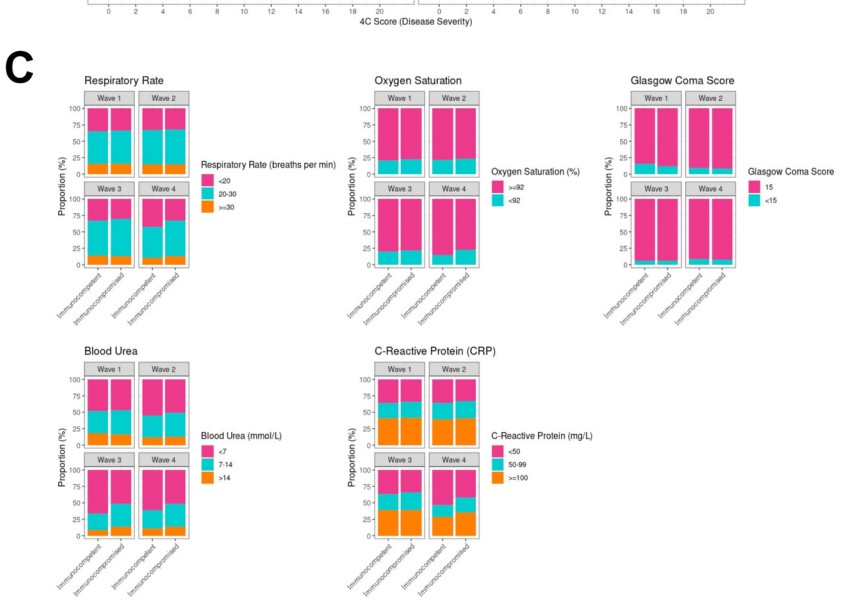

**Fig 2. Overlap between conditions giving rise to immunocompromise and illness severity.** (**A**) Overlap between immunocompromising conditions for all combinations where there are 20 patients or greater. (**B**) Illness severity over the course of the pandemic, measured using the 4C Mortality Score by immune status and pandemic wave. Pink bars: immunocompetent, green bars: immunocompromised. (**C**) Physiological components of the 4C Mortality Score stratified by immune status and pandemic wave.

## Presenting symptoms and severity of illness

Illness severity at presentation to hospital as measured by the physiological components of the 4C Mortality Score was similar in the two groups at the beginning of the pandemic, suggesting no difference in the threshold for admission for immunocompromised patients (Fig 2B). Illness severity reduced in both groups over the course of the pandemic, although to a lesser extent in the immunocompromised group (Fig 2B and 2C). When compared with immunocompetent patients, in the fourth (Omicron) wave, immunocompromised patients had a higher fraction of patients with respiratory rate $\geq$20 bpm, $SaO_2$ <92%, urea $\geq$7 mmol/l, and CRP $\geq$50 mg/l (S2 Table and Fig 2C).

## Treatments received

When restricted to patients receiving oxygen, more patients in the immunocompromised group received corticosteroids than the immunocompetent group (S3 Table and S3 Fig). For patients on oxygen with a CRP >75, more immunocompetent patients received tocilizumab compared with immunocompromised patients (S3 Table and S4 Fig).

## Outcomes

Most immunocompromised patients were more frequently admitted to critical care, and a higher proportion received both noninvasive and invasive ventilation (Table 2). Cancer patients were a notable exception, being the only group with a reduced frequency of critical care admission and invasive ventilation. In total, 6,499 of 21,954 (29%) immunocompromised patients died, compared with 28,608 of 134,598 (21%) immunocompetent patients. The highest mortality rates were seen in patients with active cancer ($n$ = 1,818 [37%]) and those on preadmission steroids ($n$ = 517 [34%]) (Table 2).

After adjustment for age, sex, ethnicity, deprivation, and comorbidities, the odds ratio (OR) for death in the immunocompromised group overall was 1.44 (95% CI [1.39, 1.5], $p$ < 0.001). There was variation in the adjusted ORs for all the outcomes across the immunocompromised groups. Critical care admission was more likely, to a similar degree, in all the immunocompromised groups, except for patients immunocompromised as a consequence of cancer treatment (OR 0.77, 95% CI [0.70, 0.85], $p$ < 0.001; Fig 3), and patients with inherited immunodeficiency, though this latter group was small. Patients on immunosuppressants (the largest group), or steroids preadmission, had consistently greater ORs for the use of critical care, noninvasive and invasive ventilation, and death. The risk of death was highest in cancer patients (OR 2.0, 95% CI [1.87, 2.25], $p$ < 0.001), followed by transplant patients (OR 1.6, 95% CI [1.39, 1.83], $p$ < 0.001), patients with inherited immunodeficiency (OR 1.41, 95% CI [1.10, 1.79], $p$ = 0.005), preadmission steroids (OR 1.47, 95% CI [1.29, 1.67], $p$ < 0.001), preadmission immunosuppressants (OR 1.24, 95% CI [1.18, 1.30], $p$ < 0.001), and HIV (OR 1.04, 95% CI [0.77, 1.37], $p$ = 0.8).

**Table 1. Demographics and comorbidities for immunocompetent and immunocompromised patients in the ISARIC WHO CCP-UK study.** The immunocompromised patients are presented as both the total, incorporating all conditions leading to immunocompromise, and by each group defined on the case record form. Data are numbers (%), except for age, which is median (IQR).

| Label | Total N | Missing N | Levels | Immunocompetent | Immunocompromised—Total | Immunocompromised—Inherited Immune Deficiency | Immunocompromised—Previous Organ Transplant | Immunocompromised—Cancer | Immunocompromised—HIV | Immunocompromised—Preadmission Immunosuppressants | Immunocompromised—Preadmission Steroids |
|---|---|---|---|---|---|---|---|---|---|---|---|
| Total N (%) | | | | 134,598 (86.0) | 21,954 (14.0) | 526 (0.3) | 1,559 (1.0) | 5,116 (3.3) | 498 (0.3) | 12,701 (8.1) | 1,554 (1.0) |
| Age on admission (years) | 156,538 (100.0) | 14 | Median (IQR) | 69.5 (53.4 to 82.0) | 71.5 (58.7 to 80.3) | 61.8 (45.3 to 77.2) | 62.4 (52.1 to 72.0) | 72.0 (62.0 to 79.7) | 57.8 (49.8 to 70.5) | 72.5 (59.1 to 81.2) | 75.1 (63.0 to 83.6) |
| | | | <50 | 27,624 (20.5) | 2,801 (12.8) | 150 (28.5) | 337 (21.6) | 400 (7.8) | 127 (25.5) | 1,618 (12.7) | 169 (10.9) |
| | | | 50–69 | 40,578 (30.1) | 7,387 (33.6) | 172 (32.7) | 761 (48.8) | 1,814 (35.5) | 243 (48.8) | 3,985 (31.4) | 412 (26.5) |
| | | | 70–79 | 26,902 (20.0) | 6,113 (27.8) | 122 (23.2) | 297 (19.1) | 1,654 (32.3) | 64 (12.9) | 3,559 (28.0) | 417 (26.8) |
| | | | 80+ | 39,482 (29.3) | 5,651 (25.7) | 82 (15.6) | 164 (10.5) | 1,248 (24.4) | 64 (12.9) | 3,537 (27.8) | 556 (35.8) |
| | | | (Missing) | 12 (0.0) | 2 (0.0) | 0 (0.0) | 0 (0.0) | 0 (0.0) | 0 (0.0) | 2 (0.0) | 0 (0.0) |
| Sex at Birth | 156,514 (100.0) | 38 | Male | 74,429 (55.3) | 11,564 (52.7) | 268 (51.0) | 964 (61.8) | 2,972 (58.1) | 283 (56.8) | 6,228 (49.0) | 849 (54.6) |
| | | | Female | 60,001 (44.6) | 10,353 (47.2) | 258 (49.0) | 593 (38.0) | 2,133 (41.7) | 214 (43.0) | 6,452 (50.8) | 703 (45.2) |
| | | | Not specified | 136 (0.1) | 31 (0.1) | 0 (0.0) | 1 (0.1) | 10 (0.2) | 1 (0.2) | 18 (0.1) | 1 (0.1) |
| | | | (Missing) | 32 (0.0) | 6 (0.0) | 0 (0.0) | 1 (0.1) | 1 (0.0) | 0 (0.0) | 3 (0.0) | 1 (0.1) |
| Ethnicity | 136,128 (87.0) | 20,424 | White | 95,260 (70.8) | 16,578 (75.5) | 308 (58.6) | 1,018 (65.3) | 4,078 (79.7) | 230 (46.2) | 9,742 (76.7) | 1,202 (77.3) |
| | | | South Asian | 7,978 (5.9) | 1,023 (4.7) | 26 (4.9) | 169 (10.8) | 139 (2.7) | 23 (4.6) | 603 (4.7) | 63 (4.1) |
| | | | Black | 3,998 (3.0) | 603 (2.7) | 95 (18.1) | 73 (4.7) | 95 (1.9) | 118 (23.7) | 194 (1.5) | 28 (1.8) |
| | | | East Asian | 730 (0.5) | 83 (0.4) | 3 (0.6) | 13 (0.8) | 16 (0.3) | 2 (0.4) | 40 (0.3) | 9 (0.6) |
| | | | Other | 8,679 (6.4) | 1,196 (5.4) | 39 (7.4) | 125 (8.0) | 261 (5.1) | 54 (10.8) | 630 (5.0) | 87 (5.6) |
| | | | (Missing) | 17,953 (13.3) | 2,471 (11.3) | 55 (10.5) | 161 (10.3) | 527 (10.3) | 71 (14.3) | 1,492 (11.7) | 165 (10.6) |
| Number of comorbidities | 156,552 (100.0) | 0 | 0 | 24,608 (18.3) | 1,153 (5.2) | 37 (7.0) | 66 (4.2) | 194 (3.8) | 1 (0.2) | 709 (5.6) | 146 (9.4) |
| | | | 1 | 27,784 (20.6) | 3,435 (15.6) | 123 (23.4) | 234 (15.0) | 915 (17.9) | 82 (16.5) | 1,832 (14.4) | 249 (16.0) |
| | | | 2+ | 82,206 (61.1) | 17,366 (79.1) | 366 (69.6) | 1,259 (80.8) | 4,007 (78.3) | 415 (83.3) | 10,160 (80.0) | 1,159 (74.6) |
| Chronic Cardiac Disease | 141,945 (90.7) | 14,607 | No | 85,390 (63.4) | 14,351 (65.4) | 372 (70.7) | 1,054 (67.6) | 3,580 (70.0) | 365 (73.3) | 8,018 (63.1) | 962 (61.9) |
| | | | Yes | 35,597 (26.4) | 6,607 (30.1) | 130 (24.7) | 439 (28.2) | 1,318 (25.8) | 107 (21.5) | 4,124 (32.5) | 489 (31.5) |
| | | | (Missing) | 13,611 (10.1) | 996 (4.5) | 24 (4.6) | 66 (4.2) | 218 (4.3) | 26 (5.2) | 559 (4.4) | 103 (6.6) |
| Hypertension | 142,801 (91.2) | 13,751 | No | 64,695 (48.1) | 11,231 (51.2) | 295 (56.1) | 618 (39.6) | 2,870 (56.1) | 265 (53.2) | 6,389 (50.3) | 794 (51.1) |
| | | | Yes | 57,117 (42.4) | 9,758 (44.4) | 206 (39.2) | 893 (57.3) | 2,049 (40.1) | 213 (42.8) | 5,731 (45.1) | 666 (42.9) |
| | | | (Missing) | 12,786 (9.5) | 965 (4.4) | 25 (4.8) | 48 (3.1) | 197 (3.9) | 20 (4.0) | 581 (4.6) | 94 (6.0) |
| Chronic Pulmonary Disease | 141,948 (90.7) | 14,604 | No | 102,408 (76.1) | 14,855 (67.7) | 387 (73.6) | 1,267 (81.3) | 3,969 (77.6) | 418 (83.9) | 7,826 (61.6) | 988 (63.6) |
| | | | Yes | 18,495 (13.7) | 6,190 (28.2) | 122 (23.2) | 229 (14.7) | 946 (18.5) | 59 (11.8) | 4,362 (34.3) | 472 (30.4) |
| | | | (Missing) | 13,695 (10.2) | 909 (4.1) | 17 (3.2) | 63 (4.0) | 201 (3.9) | 21 (4.2) | 513 (4.0) | 94 (6.0) |
| Chronic Renal Disease | 141,492 (90.4) | 15,060 | No | 101,697 (75.6) | 16,628 (75.7) | 418 (79.5) | 639 (41.0) | 4,095 (80.0) | 393 (78.9) | 9,884 (77.8) | 1,199 (77.2) |
| | | | Yes | 18,901 (14.0) | 4,266 (19.4) | 88 (16.7) | 865 (55.5) | 790 (15.4) | 79 (15.9) | 2,190 (17.2) | 254 (16.3) |
| | | | (Missing) | 14,000 (10.4) | 1,060 (4.8) | 20 (3.8) | 55 (3.5) | 231 (4.5) | 26 (5.2) | 627 (4.9) | 101 (6.5) |
| Asthma | 141,767 (90.6) | 14,785 | No | 103,088 (76.6) | 16,476 (75.0) | 424 (80.6) | 1,342 (86.1) | 4,388 (85.8) | 403 (80.9) | 8,780 (69.1) | 1,139 (73.3) |
| | | | Yes | 17,720 (13.2) | 4,483 (20.4) | 81 (15.4) | 158 (10.1) | 511 (10.0) | 73 (14.7) | 3,348 (26.4) | 312 (20.1) |
| | | | (Missing) | 13,790 (10.2) | 995 (4.5) | 21 (4.0) | 59 (3.8) | 217 (4.2) | 22 (4.4) | 573 (4.5) | 103 (6.6) |
| Liver Disease | 140,634 (89.8) | 15,918 | No | 116,080 (86.2) | 19,849 (90.4) | 471 (89.5) | 1,363 (87.4) | 4,625 (90.4) | 434 (87.1) | 11,566 (91.1) | 1,390 (89.4) |
| | | | Yes | 3,800 (2.8) | 905 (4.1) | 29 (5.5) | 121 (7.8) | 229 (4.5) | 37 (7.4) | 436 (3.4) | 53 (3.4) |
| | | | (Missing) | 14,718 (10.9) | 1,200 (5.5) | 26 (4.9) | 75 (4.8) | 262 (5.1) | 27 (5.4) | 699 (5.5) | 111 (7.1) |

(Continued)

**Table 1.** (Continued)

| Label | Total N | Missing N | Levels | Immunocompetent | Immunocompromised—Total | Immunocompromised—Inherited Immune Deficiency | Immunocompromised—Previous Organ Transplant | Immunocompromised—Cancer | Immunocompromised—HIV | Immunocompromised—Preadmission Immunosuppressants | Immunocompromised—Preadmission Steroids |
|---|---|---|---|---|---|---|---|---|---|---|---|
| Chronic Neurological Disorder | 141,119 (90.1) | 15,433 | No | 106,340 (79.0) | 18,677 (85.1) | 423 (80.4) | 1,365 (87.6) | 4,479 (87.5) | 418 (83.9) | 10,750 (84.6) | 1,242 (79.9) |
| | | | Yes | 13,957 (10.4) | 2,145 (9.8) | 79 (15.0) | 125 (8.0) | 385 (7.5) | 56 (11.2) | 1,306 (10.3) | 194 (12.5) |
| | | | (Missing) | 14,301 (10.6) | 1,132 (5.2) | 24 (4.6) | 69 (4.4) | 252 (4.9) | 24 (4.8) | 645 (5.1) | 118 (7.6) |
| Malignant Neoplasm | 140,939 (90.0) | 15,613 | No | 111,603 (82.9) | 15,739 (71.7) | 451 (85.7) | 1,276 (81.8) | 1,509 (29.5) | 439 (88.2) | 10,792 (85.0) | 1,272 (81.9) |
| | | | Yes | 8,484 (6.3) | 5,113 (23.3) | 45 (8.6) | 209 (13.4) | 3,440 (67.2) | 33 (6.6) | 1,236 (9.7) | 150 (9.7) |
| | | | (Missing) | 14,511 (10.8) | 1,102 (5.0) | 30 (5.7) | 74 (4.7) | 167 (3.3) | 26 (5.2) | 673 (5.3) | 132 (8.5) |
| Chronic Haemotologic Disease | 140,895 (90.0) | 15,657 | No | 116,431 (86.5) | 18,758 (85.4) | 351 (66.7) | 1,389 (89.1) | 3,742 (73.1) | 442 (88.8) | 11,458 (90.2) | 1,376 (88.5) |
| | | | Yes | 3,665 (2.7) | 2,041 (9.3) | 150 (28.5) | 102 (6.5) | 1,132 (22.1) | 32 (6.4) | 574 (4.5) | 51 (3.3) |
| | | | (Missing) | 14,502 (10.8) | 1,155 (5.3) | 25 (4.8) | 68 (4.4) | 242 (4.7) | 24 (4.8) | 669 (5.3) | 127 (8.2) |
| Obesity | 125,264 (80.0) | 31,288 | No | 89,557 (66.5) | 15,300 (69.7) | 367 (69.8) | 1,120 (71.8) | 3,841 (75.1) | 337 (67.7) | 8,543 (67.3) | 1,092 (70.3) |
| | | | Yes | 17,462 (13.0) | 2,945 (13.4) | 69 (13.1) | 185 (11.9) | 466 (9.1) | 80 (16.1) | 2,006 (15.8) | 139 (8.9) |
| | | | (Missing) | 27,579 (20.5) | 3,709 (16.9) | 90 (17.1) | 254 (16.3) | 809 (15.8) | 81 (16.3) | 2,152 (16.9) | 323 (20.8) |
| Diabetes | 143,343 (91.6) | 13,209 | No | 85,844 (63.8) | 15,135 (68.9) | 358 (68.1) | 950 (60.9) | 3,766 (73.6) | 324 (65.1) | 8,706 (68.5) | 1,031 (66.3) |
| | | | Yes | 36,322 (27.0) | 6,042 (27.5) | 153 (29.1) | 563 (36.1) | 1,188 (23.2) | 162 (32.5) | 3,537 (27.8) | 439 (28.2) |
| | | | (Missing) | 12,432 (9.2) | 777 (3.5) | 15 (2.9) | 46 (3.0) | 162 (3.2) | 12 (2.4) | 458 (3.6) | 84 (5.4) |
| Rheumatologic Disorder | 140,476 (89.7) | 16,076 | No | 106,762 (79.3) | 16,273 (74.1) | 411 (78.1) | 1,339 (85.9) | 4,310 (84.2) | 438 (88.0) | 8,586 (67.6) | 1,189 (76.5) |
| | | | Yes | 12,901 (9.6) | 4,540 (20.7) | 82 (15.6) | 138 (8.9) | 559 (10.9) | 33 (6.6) | 3,491 (27.5) | 237 (15.3) |
| | | | (Missing) | 14,935 (11.1) | 1,141 (5.2) | 33 (6.3) | 82 (5.3) | 247 (4.8) | 27 (5.4) | 624 (4.9) | 128 (8.2) |
| Dementia | 140,670 (89.9) | 15,882 | No | 104,563 (77.7) | 19,205 (87.5) | 458 (87.1) | 1,432 (91.9) | 4,583 (89.6) | 439 (88.2) | 11,047 (87.0) | 1,246 (80.2) |
| | | | Yes | 15,328 (11.4) | 1,574 (7.2) | 41 (7.8) | 53 (3.4) | 286 (5.6) | 29 (5.8) | 974 (7.7) | 191 (12.3) |
| | | | (Missing) | 14,707 (10.9) | 1,175 (5.4) | 27 (5.1) | 74 (4.7) | 247 (4.8) | 30 (6.0) | 680 (5.4) | 117 (7.5) |
| Malnutrition | 131,191 (83.8) | 25,361 | No | 109,531 (81.4) | 18,749 (85.4) | 443 (84.2) | 1,353 (86.8) | 4,346 (84.9) | 425 (85.3) | 10,888 (85.7) | 1,294 (83.3) |
| | | | Yes | 2,417 (1.8) | 494 (2.3) | 11 (2.1) | 34 (2.2) | 160 (3.1) | 15 (3.0) | 249 (2.0) | 25 (1.6) |
| | | | (Missing) | 22,650 (16.8) | 2,711 (12.3) | 72 (13.7) | 172 (11.0) | 610 (11.9) | 58 (11.6) | 1,564 (12.3) | 235 (15.1) |
| Smoking | 81,789 (52.2) | 74,763 | No | 35,378 (26.3) | 5,587 (25.4) | 165 (31.4) | 447 (28.7) | 1,271 (24.8) | 164 (32.9) | 3,214 (25.3) | 326 (21.0) |
| | | | Yes | 33,798 (25.1) | 7,026 (32.0) | 118 (22.4) | 347 (22.3) | 1,621 (31.7) | 109 (21.9) | 4,363 (34.4) | 468 (30.1) |
| | | | (Missing) | 65,422 (48.6) | 9,341 (42.5) | 243 (46.2) | 765 (49.1) | 2,224 (43.5) | 225 (45.2) | 5,124 (40.3) | 760 (48.9) |
| Vaccination Dose on Admission | 153,293 (97.9) | 3,259 | Unvaccinated | 114,364 (85.0) | 17,737 (80.8) | 452 (85.9) | 1,078 (69.1) | 4,077 (79.7) | 419 (84.1) | 10,267 (80.8) | 1,444 (92.9) |
| | | | First Dose | 5,112 (3.8) | 789 (3.6) | 6 (1.1) | 60 (3.8) | 166 (3.2) | 17 (3.4) | 516 (4.1) | 24 (1.5) |
| | | | Second Dose | 11,601 (8.6) | 2,804 (12.8) | 49 (9.3) | 344 (22.1) | 679 (13.3) | 41 (8.2) | 1,631 (12.8) | 60 (3.9) |
| | | | Third Dose | 592 (0.4) | 290 (1.3) | 6 (1.1) | 51 (3.3) | 92 (1.8) | 1 (0.2) | 139 (1.1) | 1 (0.1) |
| | | | Fourth Dose | 2 (0.0) | 2 (0.0) | 0 (0.0) | 0 (0.0) | 2 (0.0) | 0 (0.0) | 0 (0.0) | 0 (0.0) |
| | | | (Missing) | 2,927 (2.2) | 332 (1.5) | 13 (2.5) | 26 (1.7) | 100 (2.0) | 20 (4.0) | 148 (1.2) | 25 (1.6) |

CCP-UK, Clinical Characterisation Protocol in the United Kingdom; IQR, interquartile range; ISARIC, International Severe Acute Respiratory and emerging Infection Consortium.

**Table 2. Outcomes by reason for immunocompromise.** Data are numbers of patients (%).

| Label | Levels | Immunocompetent | Preexisting Immunological Disorder | Previous Organ Transplant | Cancer | HIV/ AIDS | Preadmission Immunosuppressants | Preadmission Steroids |
|---|---|---|---|---|---|---|---|---|
| Oxygen | No | 38,776 (29.2) | 151 (29.2) | 471 (30.6) | 1,555 (30.7) | 142 (29.0) | 2,858 (22.6) | 350 (22.7) |
|  | Yes | 94,128 (70.8) | 367 (70.8) | 1,070 (69.4) | 3,516 (69.3) | 348 (71.0) | 9,769 (77.4) | 1,195 (77.3) |
| Critical Care Admission | No | 112,738 (84.5) | 421 (81.1) | 1,209 (78.2) | 4,437 (87.3) | 367 (74.1) | 10,637 (84.1) | 1,319 (85.2) |
|  | Yes | 20,710 (15.5) | 98 (18.9) | 337 (21.8) | 648 (12.7) | 128 (25.9) | 2,005 (15.9) | 229 (14.8) |
| Noninvasive Ventilation | No | 107,620 (81.5) | 416 (80.5) | 1,170 (76.2) | 4,127 (81.8) | 369 (75.8) | 9,758 (77.7) | 1,212 (79.5) |
|  | Yes | 24,445 (18.5) | 101 (19.5) | 365 (23.8) | 918 (18.2) | 118 (24.2) | 2,796 (22.3) | 313 (20.5) |
| Invasive Ventilation | No | 123,531 (93.4) | 471 (90.9) | 1,365 (88.9) | 4,820 (95.3) | 428 (87.9) | 11,672 (92.9) | 1,415 (92.4) |
|  | Yes | 8,665 (6.6) | 47 (9.1) | 171 (11.1) | 236 (4.7) | 59 (12.1) | 898 (7.1) | 116 (7.6) |
| Death | No | 102,207 (78.1) | 391 (75.9) | 1,100 (72.8) | 3,168 (63.5) | 396 (81.6) | 8,849 (71.4) | 1,005 (66.0) |
|  | Yes | 28,608 (21.9) | 124 (24.1) | 411 (27.2) | 1,818 (36.5) | 89 (18.4) | 3,540 (28.6) | 517 (34.0) |

## Changes in mortality over time

Mortality for both immunocompetent and immunocompromised groups reduced over time. In the first wave, 10,495 of 35,261 (30%) of immunocompetent patients died, compared with 2,430 of 6,809 (36%) of immunocompromised patients. In the fourth (Omicron) wave, 726 of 6,452 (11%) immunocompetent patients died, and 153 of 815 (19%) of immunocompromised patients died (S4 Table).

After adjustment for age, sex, socioeconomic deprivation, comorbidity count, and vaccination status, compared with immunocompetent patients in wave 1, the OR for mortality remained elevated in all waves in immunocompromised patients (Fig 4). Univariable analysis showed a very similar result (S5 Fig). In the fourth wave, nearly a year after the vaccination program was initiated, the OR for death compared with the first wave was 0.38 (95% CI [0.34, 0.42], $p < 0.001$) for immunocompetent patients and 0.66 (95% CI [0.54, 0.80], $p < 0.001$) for immunocompromised patients (Fig 4). There was a decline in the number of patients enrolled in the study over time (S5 Fig).

To investigate this further, we calculated the difference in risk of death between the first and fourth waves for both immunocompetent and immunocompromised patients. Using the Bayesian framework described in the methods, the probability that this difference was less for immunocompromised patients (indicating less improvement over the course of the pandemic) was always greater than 67% (Fig 5). Across all ages, and both sexes, in-hospital mortality for immunocompromised patients improved less than for immunocompetent patients. This was particularly evident with increasing age: The probability of the reduction in hospital mortality being less for immunocompromised patients aged 50 to 69 years was 88% for men and 83% for women, and for those >80 years was 99% for men and 98% for women.

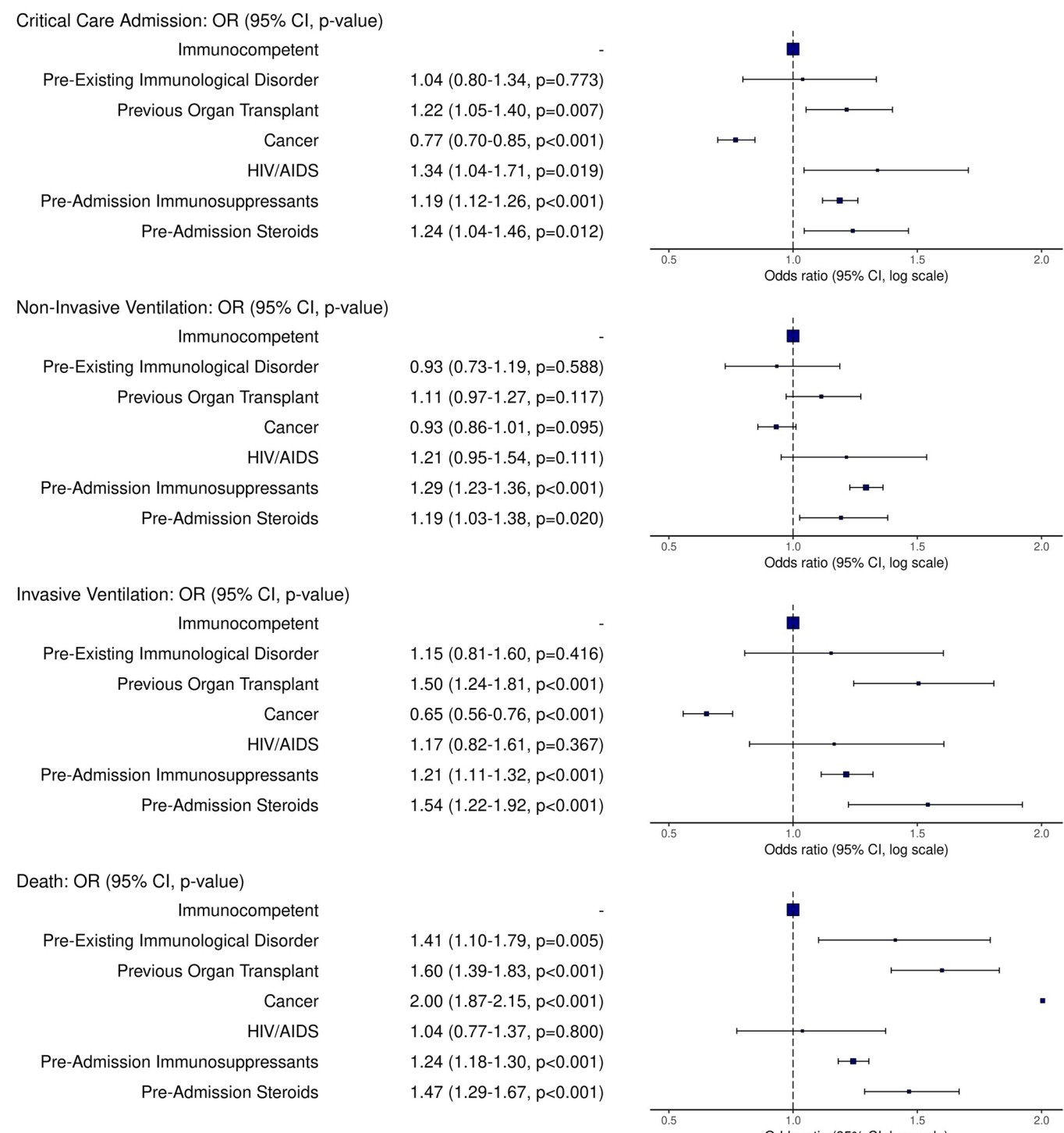

**Fig 3. Outcomes of hospitalised immunocompromised patients, compared with immunocompetent patients.** Odds ratios (ORs) from multivariable logistic regression and 95% confidence intervals (CIs) for outcomes of death, critical care admission, noninvasive and invasive ventilation, adjusted for age, sex, ethnicity, socioeconomic deprivation, chronic cardiac, pulmonary and renal disease, and vaccination status.

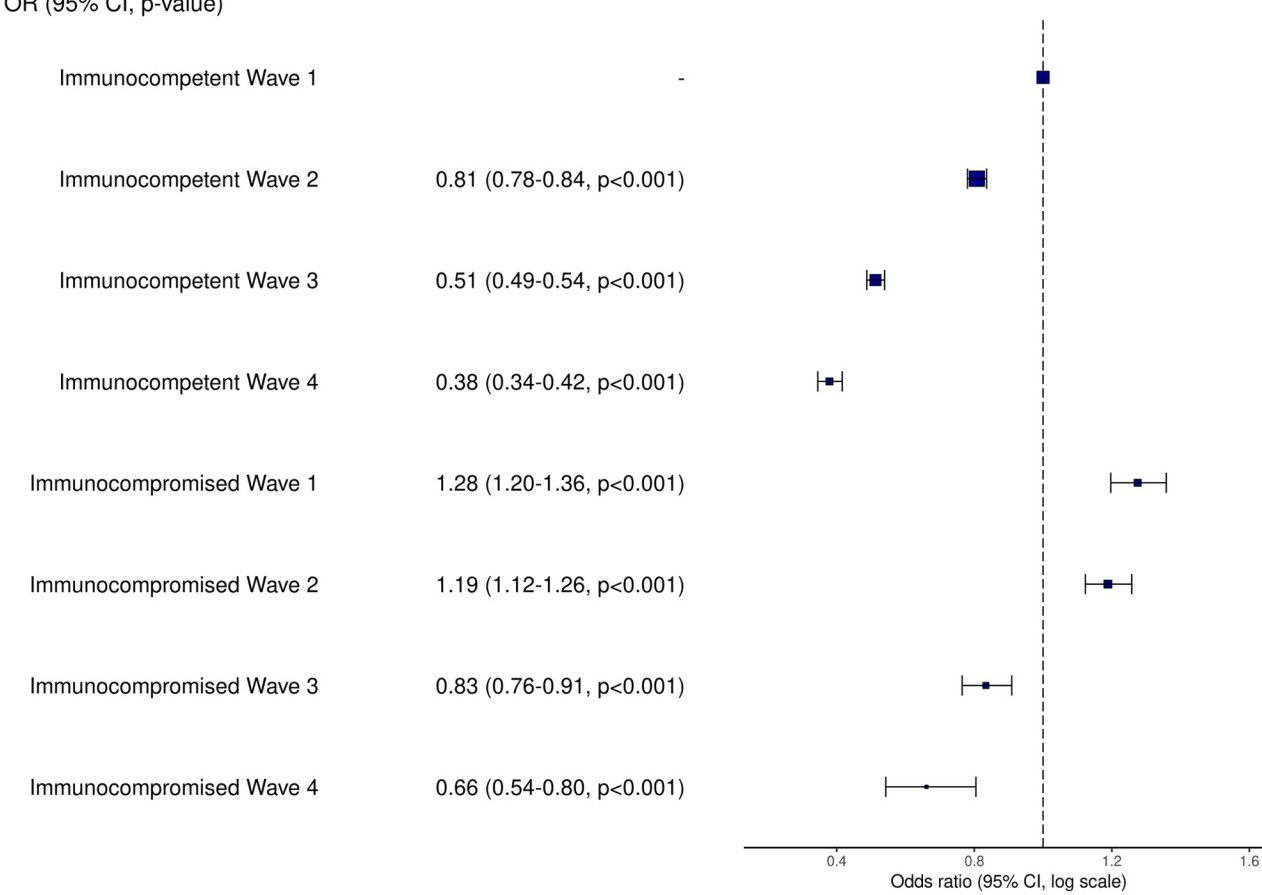

**Fig 4. Outcomes of hospitalised immunocompromised patients, compared with immunocompetent patients over the first 4 pandemic waves in the UK.** Odds ratios (ORs) from multivariable logistic regression and 95% confidence intervals (CIs) for outcomes of death, broken down by pandemic wave, adjusted for age, sex, ethnicity, socioeconomic deprivation, comorbidity count (not including the immunocompromising condition), and vaccination status. Wave 1 was 17 January to 31 August 2020; wave 2 was 1 September 2020 to 31 March 2021; wave 3 was 1 April 2021 to 12 December 2021; and wave 4 was 13 December 2021 to 28 February 2022.

### Sensitivity analysis

The immunocompromised group was less likely to be missing data than the immunocompetent group. Therefore, we conducted a sensitivity analysis using imputation of missing values to account for the effect of missing data (S6 Fig). This analysis gave a result very similar to the primary analysis shown in Fig 3. Importantly, there were no changes in significant associations when accounting for missing data. These findings did not fundamentally alter our conclusions.

## Discussion

Using the ISARIC CCP-UK cohort, we found that immunocompromised people admitted to hospital with COVID-19 had greater adjusted mortality than the general in-patient population. Over time, mortality in this group has also not fallen to the same extent as the immunocompetent patients. Over the course of the pandemic, immunocompromised patients have seen less reduction in severity at presentation to hospital and less improvement in in-hospital mortality than immunocompetent patients.

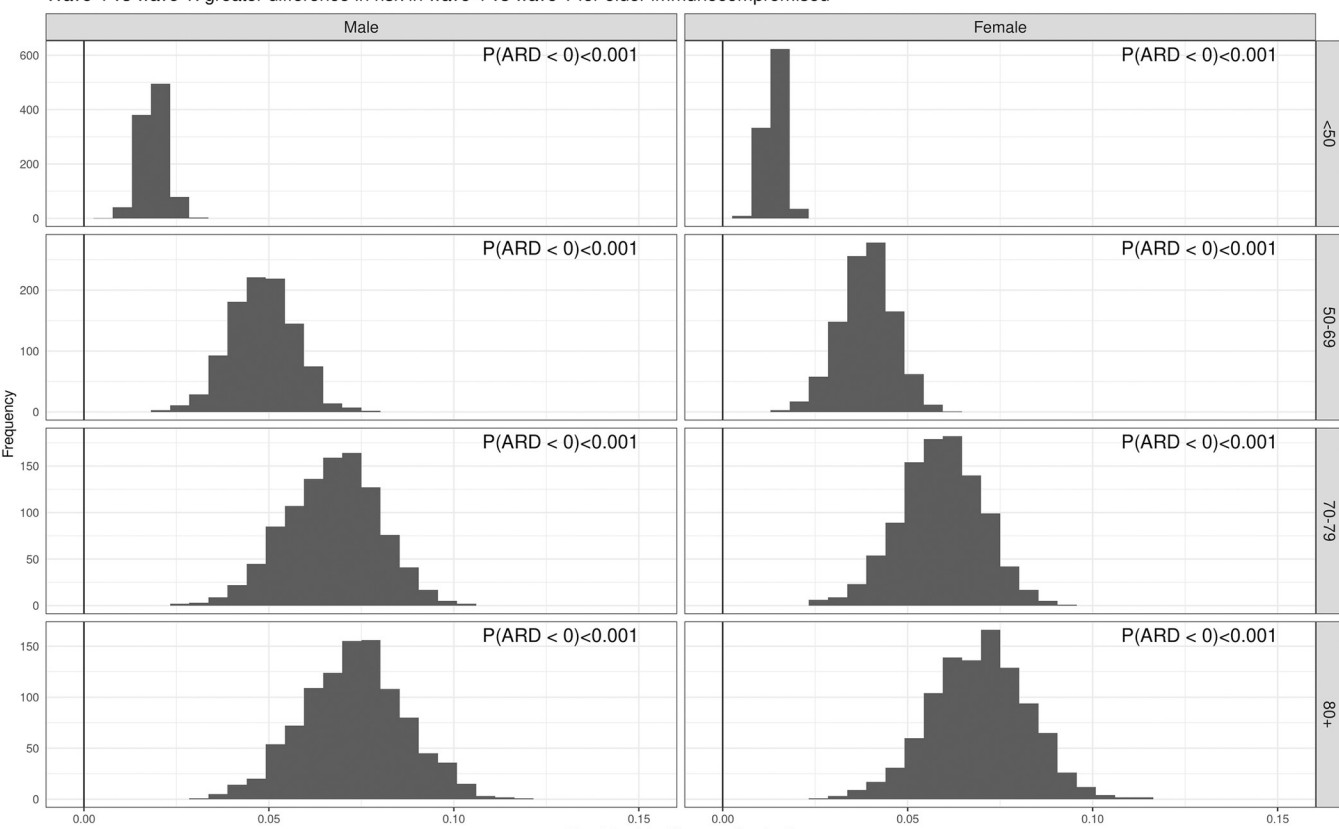

**Fig 5. Absolute risk difference (ARD) for death between immunocompromised and immunocompetent.** The difference in risk of death in the first wave and the fourth wave for immunocompetent and immunocompromised patients was modelled using Bayesian logistic regression, and the probability that the risk of death across the two waves reduced more in the immunocompetent that the immunocompromised was calculated. The analysis was stratified by age and sex.

Large population studies early in the pandemic have suggested that patients who are immunocompromised are more likely to die from COVID-19 than those who are immunocompetent [16]. However, several studies, including one of the largest to date [27], have not shown an increased risk of in-hospital mortality in immunocompromised patients when compared to those with no known immunodeficiency. This study is the only nationwide study that we are aware of that addresses this question and is therefore likely a more representative population, at least in the UK. Andersen and colleagues [27] did not find an increase in mortality in immunocompromised patients from 42 US health systems, using a propensity score matched analysis 12,841 immunocompromised and 29,386 control patients. However, the mean age of patients reported by Andersen and colleagues (59 years) was younger than our median age (71.5 years), suggesting that the difference between the two studies could be that here we are describing an older, more comorbid population. In our study, chronic pulmonary, haematologic, rheumatologic, kidney, and liver disease were all more common in immunocompromised patients, which may reflect either the reasons for immunosuppression, or complications thereof. However, comorbidities such as heart disease and hypertension were not more common in the immunocompromised patients, indicating that differences in comorbidity were not the only reason for the difference in outcomes between the groups.

One possible explanation for outcomes being the same between hospitalised immunocompromised and immunocompetent patients is that although immunocompromised patients may be more likely to be admitted to hospital, once the threshold for hospitalisation is met, the outcomes are similar. Our data challenge this hypothesis. The odds of in-hospital mortality (adjusted for age, sex, ethnicity, vaccination, and comorbidities) were higher for the immunocompromised group, with less improvement in outcome over the course of the pandemic compared with the immunocompetent group. Disease severity (physiological derangement) at presentation was similar for the two groups early in the pandemic. While severity at presentation has decreased for both groups over time, as the pandemic progressed, the improvement seen was greater for immunocompetent patients. Immunocompromised patients are now on average more ill relative to immunocompetent patients. Other possibilities for differences between studies potentially include selection of patient groups to include, criteria for hospital admission, and socioethnodemographic variation.

Immunocompromised patients with COVID-19 received steroids more frequently during their admission than immunocompetent patients. This might indicate a continuation of preadmission, non-COVID-related steroid treatment. The less frequent use of tocilizumab, on the other hand (in patients who met the criteria for use), may indicate concern about the net state of immune suppression and careful weighing of risk benefit in the minds of treating clinicians. To better inform the use of anti-inflammatory treatments in this group, future studies targeted at immunocompromised patients, or with recruitment stratified by immune status, would be needed.

There was a reduction over time in the proportion of patients admitted after vaccination compared with those unvaccinated. This difference was particularly high by the fourth wave, when most of the symptomatic patients were unvaccinated in both groups, likely reflecting the reduction in severity of the Omicron variant [28]. We have previously showed that immunocompromised patients are enriched in patients admitted after vaccination [29], although vaccination is of benefit in this patient group [30]. We cannot directly compare vaccine efficacy as we lack the denominator population, and we cannot estimate how many patients avoided admission because of vaccination. However, the risk difference between immunocompromised and immunocompetent patients widened between the first wave and the fourth wave by which time vaccination levels in the population were high. In combination with our earlier work [29], this indicates that this group of patients may remain more vulnerable than the general population even after vaccination. Selection of SARS-CoV-2 variants is largely a function of transmissibility [31,32], or immune escape [33], and a future variant could still have higher intrinsic virulence than Omicron. In this case, immunocompromised patients might be more vulnerable despite vaccination.

There are some limitations to this study. We did not collect data on the detailed drug histories of patients prior to admission, and so we did not have information on which drug was responsible for immunocompromise. We lacked data on preadmission steroid doses, meaning that we were unable to assess the extent to which steroid use contributed to immunocompromise. Missing data in the medical and drug histories means that we may have underestimated the overall effect size by categorising some immunocompromised patients as immunocompetent. This also prevented us from examining any relationship between the treatments given for COVID-19 and immunocompromise. A final weakness is that some patients early in the first wave did not have a proven diagnosis but were enrolled based on high clinical suspicion. Given that the effects we observed were present in all waves of the pandemic, we do not feel that this weakness alters our overall conclusions.

We have observed that in-hospital mortality for patients who are clinically extremely vulnerable with immunocompromise has fallen less than for immunocompetent patients in the

ISARIC CCP-UK dataset. Not all groups of immunocompromised patients had equivalent risk. The risk of death for cancer patients, in particular, was higher than for other patient groups. Cancer patients were also less likely to have their care escalated by admission to critical care or by invasive ventilation. Transplant patients also had a high risk of death, and this was despite a much higher chance of receiving interventions such as critical care admission and invasive ventilation.

A number of interventions, such as remdesivir, molnupiravir, and nirmatrelvir/ritonavir, are now available to reduce the risk of progression to severe COVID-19 and hospitalisation in this patient group [34,35]. However, some immunocompromised patients do not respond well to COVID-19 vaccines [36]. Several monoclonal antibody treatments are available for COVID-19, although their ability to neutralise is affected by viral variation [37]. Some of these antibody treatments have lost, and then regained, neutralising capacity as Omicron variants have evolved [37]. The use of such antibodies, provided they are a match for current circulating variants, as well as antiviral drugs, as early as possible, and encouraging vaccine uptake may close the gap between immunocompromised patients and the general population.

Despite the benefits of vaccination, immunocompromised patients still lag behind the general patient population in the improvements in outcomes after hospitalisation. Clinicians and policy makers should be aware of the increased risk of death in this patient group. Targeted interventions such as antiviral treatments, antibodies, and nonpharmaceutical interventions should continue to be used for immunocompromised patients with COVID-19.

## Supporting information

**S1 STROBE checklist. STROBE Checklist.**
(DOCX)

**S1 Table. Comorbidities in immunocompetent versus immunocompromised patients stratified by pandemic wave.**
(DOCX)

**S2 Table. Disease severity by immune status and pandemic wave.**
(DOCX)

**S3 Table. Number (%) of patients receiving steroids and tocilizumab by immune status and pandemic wave.**
(DOCX)

**S4 Table. Outcomes by reason for immunocompromise.**
(DOCX)

**S1 Fig. Number of admissions of immunocompetent and immunocompromised patients over time.**
(DOCX)

**S2 Fig. Admissions stratified by the number of vaccine doses received.**
(DOCX)

**S3 Fig. Steroid usage by immune status in the first 4 pandemic waves in the UK.**
(DOCX)

**S4 Fig. Tocilizumab usage by immune status in the first 4 pandemic waves in the UK.**
(DOCX)

**S5 Fig. Outcome of hospitalised immunocompromised patients, compared with immuno-competent patients—Univariable analysis.**
(DOCX)

**S6 Fig. Sensitivity analysis for the primary and secondary outcomes using imputation of missing data.**
(DOCX)

**S1 Acknowledgments. List of ISARIC4C investigators.**
(DOCX)

## Acknowledgments

We acknowledge the NIHR Clinical Research Network for providing infrastructure support for this research.

## Author Contributions

**Conceptualization:** Lance Turtle, Maaike Swets, Carlo Palmieri, Clark D. Russell, Antonia Ho, Stephen Aston, Daniel G. Wootton, Thushan I. de Silva, Peter J. M. Openshaw, Ewen M. Harrison, J. Kenneth Baillie, Malcolm G. Semple, Annemarie B. Docherty.

**Data curation:** Mathew Thorpe, Thomas M. Drake, Gary Leeming, Andy Law, Ewen M. Harrison, Annemarie B. Docherty.

**Formal analysis:** Mathew Thorpe, Thomas M. Drake, Ewen M. Harrison, Annemarie B. Docherty.

**Funding acquisition:** Lance Turtle, Carlo Palmieri, Peter J. M. Openshaw, J. Kenneth Baillie, Malcolm G. Semple.

**Investigation:** Thomas M. Drake, Maaike Swets, Clark D. Russell, Antonia Ho, Stephen Aston, Daniel G. Wootton, Alex Richter, Thushan I. de Silva, Gary Leeming.

**Project administration:** Hayley E. Hardwick, J. Kenneth Baillie, Malcolm G. Semple.

**Resources:** Andy Law.

**Software:** Andy Law.

**Supervision:** Peter J. M. Openshaw, J. Kenneth Baillie, Malcolm G. Semple, Annemarie B. Docherty.

**Validation:** Thomas M. Drake, Gary Leeming, Andy Law, Ewen M. Harrison.

**Visualization:** Mathew Thorpe, Thomas M. Drake.

**Writing – original draft:** Lance Turtle, Maaike Swets, Clark D. Russell, Antonia Ho, Stephen Aston, Daniel G. Wootton, Thushan I. de Silva, Annemarie B. Docherty.

**Writing – review & editing:** Lance Turtle, Thomas M. Drake, Maaike Swets, Carlo Palmieri, Clark D. Russell, Antonia Ho, Stephen Aston, Daniel G. Wootton, Alex Richter, Thushan I. de Silva, Hayley E. Hardwick, Gary Leeming, Andy Law, Peter J. M. Openshaw, Ewen M. Harrison, J. Kenneth Baillie, Malcolm G. Semple, Annemarie B. Docherty.

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
