## [Editor Report · Decision Letter 0]

4 Aug 2022

Dear Dr Turtle, 

Thank you for submitting your manuscript entitled "Outcome of COVID-19 in hospitalised immunocompromised patients: an analysis of the WHO ISARIC CCP-UK prospective cohort study." for consideration by PLOS Medicine.

Your manuscript has now been evaluated by the PLOS Medicine editorial staff as well as by an academic editor with relevant expertise and I am writing to let you know that we would like to send your submission out for external peer review.

Please re-submit your manuscript within two working days, i.e. by Aug 08 2022 11:59PM.

Kind regards,

Philippa

Dr. Philippa Dodd, MBBS MRCP PhD

Senior Editor

PLOS Medicine

---

## [Decision Letter · Decision Letter 1]

28 Sep 2022

Dear Dr. Turtle,

Thank you very much for submitting your manuscript "Outcome of COVID-19 in hospitalised immunocompromised patients: an analysis of the WHO ISARIC CCP-UK prospective cohort study." (PMEDICINE-D-22-02623R1) for consideration at PLOS Medicine. 

[LINK]

In light of these reviews, I am afraid that we will not be able to accept the manuscript for publication in the journal in its current form, but we would like to consider a revised version that addresses the reviewers' and editors' comments. Obviously we cannot make any decision about publication until we have seen the revised manuscript and your response, and we plan to seek re-review by one or more of the reviewers. 

We expect to receive your revised manuscript by Oct 13 2022 11:59PM. Please email us (plosmedicine@plos.org) if you have any questions or concerns.

We look forward to receiving your revised manuscript. 

Sincerely,

Philippa Dodd, MBBS MRCP PhD

PLOS Medicine

pdodd@plos.org

plosmedicine.org

PLOS Medicine requires that the de-identified data underlying the specific results in a published article be made available, without restrictions on access, in a public repository or as Supporting Information at the time of article publication, provided it is legal and ethical to do so. Please see the policy at http://journals.plos.org/plosmedicine/s/data-availability and FAQs at http://journals.plos.org/plosmedicine/s/data-availability#loc-faqs-for-data-policy

For each data source used in your study: 

Did your study have a prospective protocol or analysis plan? Please state this (either way) early in the Methods section. 

ABSTRACT

Please structure your abstract using the PLOS Medicine headings (Background, Methods and Findings, Conclusions). Please combine the Methods and Findings sections into one section, “Methods and findings”.

Abstract Methods and Findings: 

Please ensure that all numbers presented in the abstract are present and identical to numbers presented in the main manuscript text.

Please include the study design i.e. prospective, observational), length of follow up, and main outcome measures.

Please include the important dependent variables that are adjusted for in the analyses. 

In the last sentence of the Abstract Methods and Findings section, please describe the main limitation(s) of the study's methodology.

Abstract Conclusions:

Line 80: “...additional vaccine doses and monoclonal antibodies” any role for non-pharma preventative interventions (masks, distancing in-hospitals etc)? might be worth a mention

METHODS and RESULTS

Please define "lost to follow-up" as used in this study. Other reasons for exclusion should be defined.

Please define the length of follow up (eg, in mean, SD, and range).

Line 270: “…(median age 72.0. IQR (62.0 to 79.7))” please amend the presentation of these data to the following: (median age 72.0 [IQR 62.0-79.7]) and throughout the manuscript

Similarly line 285: (n=6,607 (30%)) � (n=6,607 [30%]) and throughout

Line 301: Please provide the actual numbers of events for the outcomes, not just summary statistics , %, or ORs. Please check and amend throughout

Line 341: please present numerators and denominators for percentages – please check and amend throughout

For the following analyses: outcomes of hospitalised immunocompromised patients, compared with immunocompetent patients presented in figure 2 please also provide unadjusted anlayses

FIGURES

Please provide titles and legends for all figures (including those in Supporting Information files).

Figure 2 – as above, please also present unadjusted analyses (you may wish to consider the use of a table)

TABLES

Table S4: Please define numbers and those in brackets

SUPPLEMENTARY INFORMATION

Please provide titles AND legends for each individual table and figure in the Supporting Information.

You have included a CONSORT diagram (figure S1) which would be more relevant for an RCT. The information contained within is helpful to the reader and I have no objection to it remaining but would suggest a revised title and an appropriately associated legend

Please ensure that the study is reported according to the STROBE guideline and include the completed STROBE checklist as Supporting Information. Please add the following statement, or similar, to the Methods: "This study is reported as per the Strengthening the Reporting of Observational Studies in Epidemiology (STROBE) guideline (S1 Checklist)." The STROBE guideline can be found here: http://www.equator-network.org/reporting-guidelines/strobe/ When completing the checklist, please use section and paragraph numbers, rather than page numbers.

REFERENCES

Please select the PLOS Medicine reference style in your citation manager. In-text reference call outs should be presented as follows noting the absence of spaces within the square brackets, “…symptomatic [2,8].”

Please ensure no more than 6 author names are listed in the bibliography before et al (if there are more than 6 contributing authors)

Comments from the reviewers:

Reviewer #1: This work evaluates the impact of immunosuppression on the outcome of COVID-19 patients admitted to the hospital using the large WHO ISARIC CCP-UK cohort involving the four waves of the pandemic in Wales and England. This is a very relevant issue with important potential consequences in the design of preventive policies to protect this specific collective. Results from this study support that the presence of a basal status of immunosuppression is a risk factor for hospital mortality and to lesser extent to other outcomes indicating severity (receiving oxygen, critical care admission, non-invasive ventilation and invasive ventilation). Although the mortality decreased with the pandemic course, this decrease was less patent in those patients with immunosuppression. This reviewer has some comments/suggestions.

1. In the methods section, the authors state that "To examine in-hospital mortality, we initially performed multivariable logistic regression adjusted for age, sex, number of comorbidities, ethnicity, deprivation index and vaccination". Nonetheless, in the Fig 2b legend, the authors sate that "ORs are adjusted for age, sex, ethnicity, deprivation, and comorbidities". Why vaccination is gone here?

2. I wonder if the association between immunosuppression and mortality is still present after adjustment by the pharmacological treatments received during hospitalization: steroids, tocilizumab, remdesivir…If this analysis is not feasible, I suggest acknowledging it as a limitation. 

3. Comorbidities are introduced in the model as "number of comorbidities", instead of as individual entities. Please justify it. If an aggregated score is used, is it possible to use the Charlson score?

4. In the methods section, the authors state that during the first wave, highly suspected cases were also eligible for inclusion, because SARS-CoV-2 was an emergent pathogen at that time and laboratory confirmation was dependent on availability of testing. Are the results the same excluding these patients from the analysis, I mean, maintaining only those with a confirmed SARS-CoV-2 infection? A recent study has demonstrated that while COVID-19 disease was a large component of total aLRTD during the pandemic period, non- SARS-CoV-2 infection still caused the majority of respiratory infection hospitalisations (DOI:https://doi.org/10.1016/j.lanepe.2022.100473)

5. It would be interesting to analyze if vaccination was a protective factor for those patients admitted to the hospital with an antecedent of immunosuppression, comparing the outcome of vaccinated versus unvaccinated immunosuppressed patients 

Reviewer #2: Turtle et al present an analysis from a large national prospective COVID-19 cohort. The large number of patients allows for meaningful sub-group analysis. Only patient data sets linkable to the national immunization management system were analyzed. Two categories were compared: Immunosuppressed, defined by diagnosis (e.g. HIV) and use of immunosuppressive drugs (e.g. steroids pre-admission), with "non"-immunosuppressed patients. The adjusted analysis (with variables commonly associated with worse outcome of COVID-19) shows a remaining elevated risk of death from COVID-19 for immunocompromised patients. This risk, however, is decreasing over time.

The authors conclude that targeted measures such as additional boosters and monoclonal antibodies should be considered.

With all its biases, it is one of the largest descriptive studies. A number of remarks/ issues are detailed below.

Concerns/remarks

Even if HIV is counted in the immunocompromised column, it is debatable whether a well-treated HIV patient should be considered immunocompromised. I would be helpful to conduct, if reasonable, some sensitivity analysis with HIV patient alone or without including them into the immunocompromised group.

In the conclusion, it would be important to mention "early "therapy with mAB but DAA as well, and prophylaxis, if working with the circulating variant.

Reviewer #3: See attachment

Michael Dewey

[LINK]

---

## [Decision Letter · Decision Letter 2]

20 Dec 2022

Dear Dr. Turtle,

Thank you very much for re-submitting your manuscript "Outcome of COVID-19 in hospitalised immunocompromised patients: an analysis of the WHO ISARIC CCP-UK prospective cohort study." (PMEDICINE-D-22-02623R2) for review by PLOS Medicine.

I have discussed the paper with my colleagues and the academic editor and it was also seen again by 3 reviewers. I am pleased to say that provided the remaining editorial and production issues are dealt with we are planning to accept the paper for publication in the journal.

[LINK]

We look forward to receiving the revised manuscript by Dec 27 2022 11:59PM.   

Sincerely,

Philippa Dodd, MBBS MRCP PhD

PLOS Medicine

plosmedicine.org

Requests from Editors:

GENERAL

Thank you for your detailed responses to previous editor and reviewer comments

COMMENTS FROM THE ACADEMIC EDITOR

I would particularly like to see the authors include some of their explanations to the reviewers as text in the article itself. They might have done some of this, but since they omit line numbers in many of their comments, I can't tell.

1. Response to Reviewer #1, comment 2 (adjustment for pharmacological treatment) please sign post to where this is addressed

2. They say they mean to start from age 19 because they already published the paediatric data. Please state this, under 'Participants' and cite the publication (see comments below also)

3. Reviewer #2, 'Upset plot' - can they delete 'Upset'? I don’t know what that is and I don't think it's helpful

4. Reviewer #2, S1 Table – regarding the proportion of unvaccinated people – please add an appropriate explanation to the discussion. 

They have dealt with all my own comments.

** In addressing point 4 above, the editorial team suggest that you take care with your wording as the explanation in your rebuttal (without the question) could be confused with justification simply for the number of number of admissions i.e. please be sure to refer to those ppts as unvaccinated

ABSTRACT

In the last sentence of the methods and findings section please include the main limitations of your methodology.

Please remove the funding statement from the end of the abstract and include only in the submission form. 

AUTHOR SUMMARY

Thank you for including an author summary. Please structure as follows, separating each paragraph into (1-4) individually bulleted points below each sub-heading:

Why Was This Study Done? 

Authors should reflect on what was known about the topic before the research was published and why the research was needed.

What Did the Researchers Do and Find? 

Authors should briefly describe the study design that was used and the study’s major findings. Do include the headline numbers from the study, such as the sample size and key findings. 

What Do These Findings Mean? 

Authors should reflect on the new knowledge generated by the research and the implications for practice, research, policy, or public health. Authors should also consider how the interpretation of the study’s findings may be affected by the study limitations.

INTRODUCTION 

Line 146: please remove the word “differed” it is redundant here

METHODS and RESULTS

Line 374 (for example): suggest the following - “(OR 2.0, 95% CI [1.87,2.25], p<0.001)” note the addition of parentheses around CIs and suggest considering the use of a comma instead of a hyphen (hyphens can become confused with reporting of negative values). Please check and amend throughout, including in the abstract

Reviewer #3 response to age inclusion (18 Vs 19 year olds) “paediatric analyses included all patients up to and including the age of 18 years”– thank you for clarifying this point. It would be helpful to add this clarification to the appropriate part of the manuscript we suggest including this under the sub-heading “participants”

DISCUSSION

Please present and organize the Discussion as follows: a short, clear summary of the article's findings; what the study adds to existing research and where and why the results may differ from previous research; strengths and limitations of the study; implications and next steps for research, clinical practice, and/or public policy; one-paragraph conclusion. 

The component parts all seem to be there, but a little re-structuring and a clearly defined conclusion would be of benefit

Please remove the data availability statement from the end of the discussion (line 554) and include only in the submission form, it will be compiled as meta-data.

REFERENCES

Line 131 (for example): “[13, 14].” Please remove spaces between citations as follows: “[13,14]”.

Please add reference to the paediatric data already published which includes 18 year olds (see also academic editor comment above)

SOCIAL MEDIA

To help us extend the reach of your research, please provide any Twitter handle(s) that would be appropriate to tag, including your own, your co-authors’, your institution, funder, or lab. Please respond to this email with any handles you wish to be included when we tweet this paper. Please include these in the submission form when you re-submit your manuscript

Comments from Reviewers:

Reviewer #1: The authors have appropirately addressed the points raised in my previous report. This is an important article confirming immunosupressed as a group at risk of poor outcomes following infection by SARS-CoV-2. These results support those interventions aimed at preventing uncontrolled viral replciation in these patients (vacccines, early antiviral treatment, (active) monoclonal antibodies)

Reviewer #2: The authors have carefully responded to all concerns raised, and improved the manuscript.

Reviewer #3: The authors have addressed all my points.

Michael Dewey

[LINK]

---

## [Editor Report · Decision Letter 3]

11 Jan 2023

Dear Dr Turtle, 

On behalf of my colleagues and the Academic Editor, Professor Nicola Low, I am pleased to inform you that we have agreed to publish your manuscript "Outcome of COVID-19 in hospitalised immunocompromised patients: an analysis of the WHO ISARIC CCP-UK prospective cohort study." (PMEDICINE-D-22-02623R3) in PLOS Medicine.

Please make the following amendments to prior to publication. Please see below:

AUTHOR SUMMARY - please ensure each statement is bullet pointed

Please split the sentence at line 95 where it begins “Over time…” such that this is a separate bullet point

Please split the sentence at line 105 where it begins “Our analysis…” such that this is a separate bullet point

Line 98: the sentence is long and in the last part not entirely clear, please revise for clarity of this point. In addition, “However…” follows a comma please correct to a lower-case h

Please split the sentence at line 114 where it begins “Clinicians…” such that this is a separate bullet point

Please split the sentence at line 115 where it begins “Targeted interventions…” such that this is a separate bullet point

PRESS

Best wishes,

Pippa

Philippa Dodd, MBBS MRCP PhD 

PLOS Medicine